# A Fully Differential Difference Transconductance Amplifier Topology Based on CMOS Inverters

Otávio Soares Silva [1,*], Rodrigo Aparecido da Silva Braga [1,*], Paulo Marcos Pinto [2], Luís Henrique de Carvalho Ferreira [2] and Gustavo Della Colletta [2]

1 Institute of Science and Technology, Federal University of Itajuba, Itabira 35903-087, MG, Brazil
2 Institute of Systems Engineering and Information Technology, Federal University of Itajuba, Itajuba 37500-903, MG, Brazil
* Correspondence: otaviosoaressilva@unifei.edu.br (O.S.S.); rodrigobraga@unifei.edu.br (R.A.d.S.B.)

**Abstract:** This manuscript presents a fully differential difference transconductance amplifier (FDDTA) architecture based on CMOS inverters. Designed in a 130-nm CMOS process it operates in weak inversion when supplied with 0.25 V. In addition, the FDDTA requires no supplementary external calibration circuit, like tail current or bias voltage sources, since it relies on the distributed layout technique that intrinsically matches the CMOS inverters. For analytical purposes, we carried out a detailed investigation that describes all the concepts and the whole operation of the FDDTA architecture. Furthermore, a comparison between the modeling equations and measured data assures high performance.

**Keywords:** fully differential difference transconductance amplifier; CMOS inverters; differential buffer configuration; weak inversion region; low-power circuits

## 1. Introduction

As CMOS processes continue to develop, the demand for power reduction and lower supply voltage becomes more apparent. In some instances, such reductions may provide smaller devices like implantable chips, mobile phones, IoT electronic sensors, portable medical devices, etc.

Furthermore, since a majority of this equipment is made up of analog and digital blocks [1], which are embedded by the MOS transistor, shrinking the transistor's size also lowers supply voltage. This reduction is mainly due to the transistor's operating region [2], and, therefore, enables mobile devices to become more independent from recharging sources for a longer time and allows for more effective and safe use of the battery.

Power supply reduction can help to lower energy consumption according to recent literature [3–6], but analog block degradation of dynamic range (DR) occurs [7]. A good alternative is to use electronic blocks that alter differential signals to mitigate the loss in DR. Differential signal processing is superior, in terms of dynamic range and power supply rejection, when compared with single-ended processes, and differential signals can, thereby, eliminate common-mode noises and disturbances [8].

In our previous work, we studied the topology of Nauta OTA [9] adapted to operate in ultra-low power and low displacement voltage, using arrays of halo-implanted transistors [10]. Based on this work, we developed an FDDTA that was used in a fifth-order Butterworth low-pass filter [11]. As a complement to this research, this paper carries out a complete characterization of the FDDTA used in [11].

The analog building block widely employed to handle differential signals is the differential voltage amplifier, the output of which is proportional to the difference between two voltage inputs. Operational transconductance amplifiers (OTAs) usually consist of three stages: a common mode rejection stage, that rejects input variations; a gain stage, that amplifies the signal; and a driver stage, that provides output resistive load [12]. They

are also capable of manipulating differential signals. However, they output a differential current signal.

Among the possible applications of OTAs include Gm-C filters, in which biomedical applications, having frequency ranges which vary below 100 Hz, can be highlighted [13]. In a similar application area, the fully differential DIGOTA [14] is a biomedical application, which combines a Muller C-element with a tri-state buffer to allow FD operation.

Another class of differential amplifiers is the differential difference amplifier (DDA). Proposed by Säckinger and Guggenbuhl [15], the DDA is an extension of the operational amplifier concept. Differing from the op-amp idea, the DDA compares two differential signals, and its fully-differential version requires a common-mode control circuit, similar to single-ended amplifiers.

The availability of multiple inputs makes this amp attractive for many applications, such as the following: filters, for example, ref. [16] presented a $G_m$-C filter application employing low-power DDA; transconductance amplifiers, utilizing a common-mode feedback circuit suitable for fully balanced analog MOS structures, as displayed in the work by [17]; self-adaptive power consumption microphone preamplifiers, as proposed in [18]. DDA circuits have been highly studied in the past. In 1987, ref. [15] published a paper concerning a DDA implemented in a double-poly CMOS technology, featuring two differential inputs. In 1994, ref. [19] presented a DDA amplifier utilizing the body effect to improve linearity. In 2001, a low-power wide input range was presented in the work of [20]. Despite all this research, nowadays. little attention is dedicated to the architectural level.

Herein, we designed a fully differential difference transconductance amplifier (FDDTA) to reduce noise and power consumption at the system architectural level. This amplification technique translates into voltage supply reduction.

The FDDTA was designed in a 130 nm CMOS and operated efficiently in weak inversion when supplied with 0.25 V. The FDDTA did not require an external calibration circuit, like a tail current or bias voltage source, since it was based on the distributed layout technique, which inherently matched the CMOS inverters. Furthermore, we constructed a completely-differential buffer configuration for validation purposes.

The remainder of this manuscript is organized as follows. Section 2 elucidates the background theory. Section 3 provides the proposed FDDTA topology and concepts. The measurements are presented in Section 4. Finally, Section 5 concludes our contributions.

## 2. Materials and Methods

### 2.1. The Conceptual FDDTA

The FDDTA, illustrated in Figure 1, is a six-terminal device that comprises two differential voltage input ports, ($V_{pp} - V_{pn}$) and ($V_{np} - V_{nn}$), and a differential output stage ($I_{op} - I_{on}$). Operating in the linear range, the output is:

$$I_{od} = I_{op} - I_{on} = G_m[(V_{pp} - V_{pn}) - (V_{np} - V_{nn})], \qquad (1)$$

where $G_m$ states the small signal transconductance of the FDDTA.

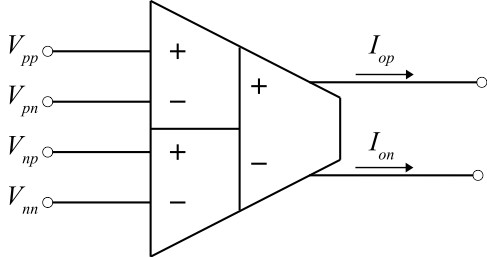

**Figure 1.** The FDDTA symbol, comprised of two differential voltage input ports and a differential output stage.

Figure 2 illustrates the FDDTA buffer configuration. comparing it to a single-ended OTA buffer configuration, and, thereby, showing that both follow the same feedback principle.

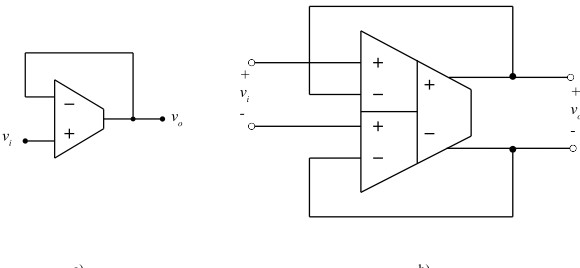

**Figure 2.** A fundamental application case: (**a**) OTA buffer configuration (single-ended). (**b**) FDDTA buffer configuration (fully-differential).

### 2.2. Weak Inversion Operation

The drain current $I_{DS}$ of a long channel MOS transistor operating in weak inversion is based on the channel diffusion current according to:

$$I_{DS} = I_{D0}\left(\frac{W}{L}\right)\exp\left(q\frac{V_{GS}}{nkT}\right)\left[1 - \exp\left(-q\frac{V_{DS}}{kT}\right)\right],\tag{2}$$

where $I_{D0}$ (physical and process parameters) is the minimum drain current and $n$ the slope factor in weak inversion. All the other symbols have their usual meanings. In addition, the transistor is saturated when ($V_{DS} \geq 3kT/q$) [2], which translates into lower supply voltage.

### 2.3. CMOS Inverter

The weak inversion operation is an effective way to reduce power consumption, something that, given [21], suits our design specifications well, since the proposed FDDTA contains a number of inverter blocks. Consequently, it was essential to study the CMOS inverter functioning in weak inversion prior to expanding these ideas to the entire circuit.

#### 2.3.1. Transconductance of the CMOS Inverter

The circuit of Figure 3 illustrates the schematic of a CMOS inverter's basic cell. Considering all transistors saturated in weak inversion ($V_{DS} \geq 3kT/q$), and applying (2), we obtain $I_p$ and $I_n$ [10]

$$I_p = I_{D0_p}\left(\frac{W}{L}\right)_p\exp\left(q\frac{V_{DD} - V_i}{nkT}\right)\tag{3a}$$

and

$$I_n = I_{D0_n}\left(\frac{W}{L}\right)_n\exp\left(q\frac{V_i}{nkT}\right).\tag{3b}$$

Assuming $V_i = V_o$, both pMOS and nMOS transistors conduct the same short circuit current, $I_{SC}$. This current charges the inverter up to operate at its threshold voltage, $V_{SP}$ [9]. Since a single inverter works as an amplifier when biased around the point $V_i = V_o$ [8], we calculate $I_{SC}$ for $V_{SP} = V_{DD}/2$, according to:

$$I_{SC} \overset{\triangle}{=} I_{D0_p}\left(\frac{W}{L}\right)_p\exp\left(q\frac{V_{TH}}{nkT}\right) = I_{D0_n}\left(\frac{W}{L}\right)_n\exp\left(q\frac{V_{TH}}{nkT}\right).\tag{4}$$

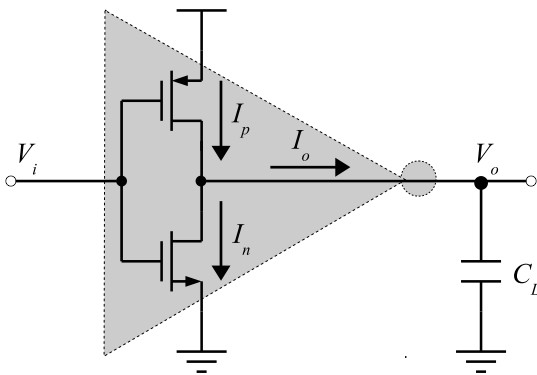

**Figure 3.** CMOS inverter basic cell: schematic and symbols.

Note that we can establish the threshold voltage by choosing appropriate transistor geometries and also design the $I_{SC}$ current [10]. In addition, from Figure 3, the output current is calculated as $I_o = I_p - I_n$, and by invoking (4), we have:

$$I_o = 2I_{SC} \sinh\left( q\frac{V_{TH} - V_{in}}{nkT} \right). \tag{5}$$

Consequently, differentiating $I_o$ should then give the effective transconductance of the CMOS inverter at the bias point $V_i = V_{TH}$, according to:

$$\left.\frac{\partial I_o}{\partial V_i}\right|_{V_i=V_{TH}} = -2q\frac{I_{SC}}{nkT} = -(g_{m_p} + g_{m_n}), \tag{6}$$

where $g_{m_p}$ and $g_{m_n}$ are the transconductances of the pMOS and nMOS transistors, respectively. At this point, we define $G_m \triangleq g_{m_n} + g_{m_p}$ and $G_o$ is the sum $g_{o_p} + g_{o_n}$ to simplify further equations. In other words, they neither depend on biasing nor geometry parameters, since they are functions of physical parameters [2,10].

### 2.3.2. Small-Signal AC Model

The small-signal AC equivalent circuit model of the CMOS inverter has the following transfer function:

$$\frac{v_o(s)}{v_i(s)} = -\frac{g_{m_p} + g_{m_n}}{sC_L + g_{o_p} + g_{o_n}} = \frac{-G_m}{(sC_L + G_o)}, \tag{7}$$

where $sC_L$ incorporates the parasitic capacitances inherent to the circuit and the capacitive load. In addition, $g_{o_p}$ and $g_{o_n}$ are the output conductances of the pMOS and nMOS transistors, respectively.

### 3. Results

As illustrated in Figure 4, the proposed FDDTA was comprised of eight CMOS inverters. When both pMOS and nMOS transistors were intrinsically matched, a more linear CMOS V-I conversion was achieved [10], thus reducing distortion effects.

The input stage is characterized by inverters INV1-INV4. All others, INV5 to INV8, are responsible for controlling $I_{op}$ and $I_{on}$ outputs. The cross-connected inverters, INV7 and INV8, inject currents in the impedances represented by the self-connected inverters, INV5 and INV6, respectively. This proposed schematic was based on previous work developed by [9], employed for integrated analog filters at very high frequencies, based on transconductance-C integrators. This architecture requires no auxiliary external calibration circuit, such as tail current or bias voltage, sources.

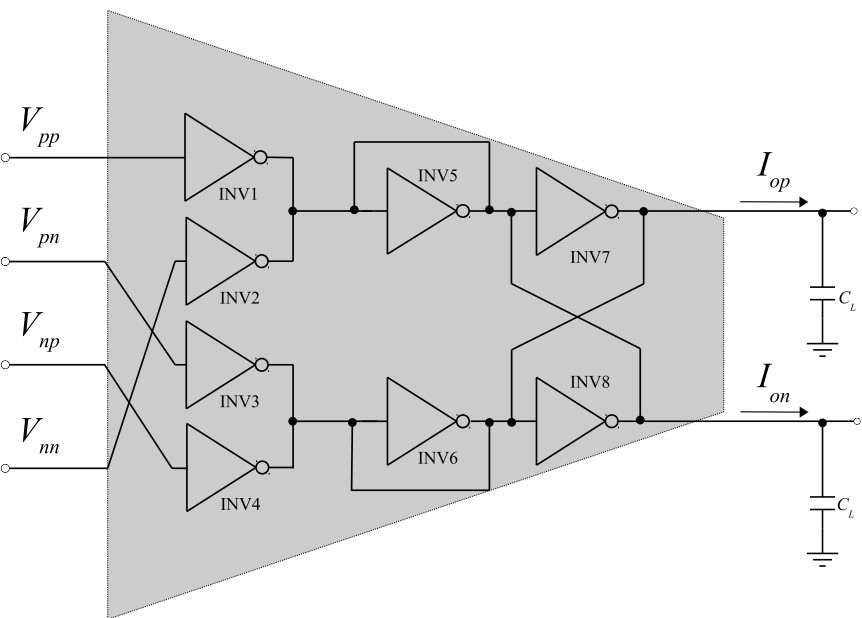

**Figure 4.** The proposed fully-differential difference amplifier schematic.

### 3.1. The FDDTA

In this subsection, we describe the modeling of the entire FDDTA circuit based on the previous concepts. In addition, we cover the overall design and the small-signal AC inherent model.

#### 3.1.1. Transconductance of the FDDTA

Taking into account the circuit shown in Figure 4, where two differential signals ($V_{id1}$, $V_{id2}$) are applied to the FDDTA inputs, we can write:

$$V_{pp} = V_{SP} + \frac{V_{id1}}{2} \quad , \quad V_{pn} = V_{SP} - \frac{V_{id1}}{2}; \tag{8a}$$

and

$$V_{np} = V_{SP} + \frac{V_{id2}}{2} \quad , \quad V_{nn} = V_{SP} - \frac{V_{id2}}{2}. \tag{8b}$$

Regarding the fact that all transistors are similar, we obtain the differential output current, $I_{od} = I_{op} - I_{on}$, by invoking (5). Regarding the switching point, $V_{SP} = V_{DD}/2$, for all CMOS inverters [9], we obtain:

$$I_{od} = 4I_{SC}\left[\sinh\left(q\frac{V_{id1}}{2nkT}\right) - \sinh\left(q\frac{V_{id2}}{2nkT}\right)\right]. \tag{9}$$

Expanding (9) into Taylor series, around $V_{SP}$, leads to:

$$I_{od} = 2q\frac{I_{SC}}{nkT}(V_{id1} - V_{id2}) = G_m[(V_{pp} - V_{pn}) - (V_{np} - V_{nn})], \tag{10}$$

as required by (1) to be a FDDTA.

#### 3.1.2. Small-Signal AC Model

We establish that $g_{m_i} = g_{m_{p_i}} + g_{m_{n_i}}$ and $g_{o_i} = g_{o_{p_i}} + g_{o_{n_i}}$; leading to the small-signal model depicted in Figure 5, having output voltages ($v_{o_p}$) and ($v_{o_n}$):

$$v_{o_n}(s) = -\frac{g_{m_1}v_{pp}(s) + g_{m_2}v_{nn}(s) + g_{m_8}v_{o_p}(s)}{sC_L + (g_{o_1} + g_{o_2} + g_{o_5} + g_{o_8} + g_{m_5})} \tag{11a}$$

and

$$v_{o_p}(s) = -\frac{g_{m_3}v_{pn}(s) + g_{m_4}v_{np}(s) + g_{m_7}v_{o_n}(s)}{sC_L + (g_{o_3} + g_{o_4} + g_{o_6} + g_{o_7} + g_{m_6})}. \tag{11b}$$

Manipulating (11a) and (11b), and regarding the same $g_m$ and $g_o$ for all transistors, results in a differential output signal according to:

$$\frac{[v_{o_p}(s) - v_{o_n}(s)]}{[v_{pp}(s) - v_{pn}(s)] - [v_{np}(s) - v_{nn}(s)]} = \frac{G_m}{sC_L + 4G_o}, \tag{12}$$

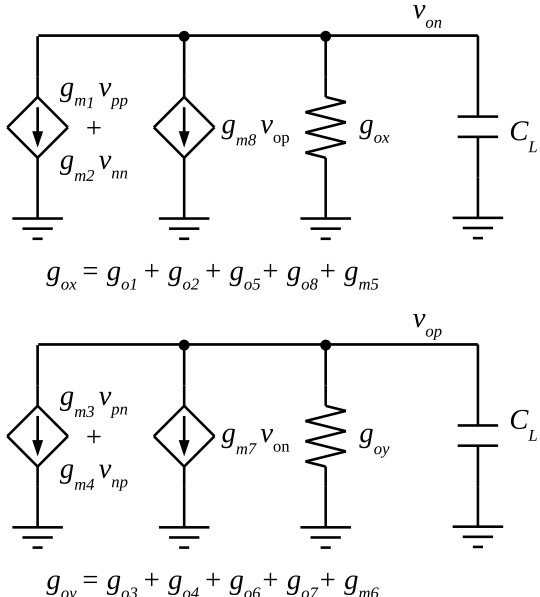

**Figure 5.** The small-signal AC model of the FDDTA.

## 4. Discussion

Since the overall design follows prior work developed in [10] we used a similar ($8 \times 8$) array of unity halo-implanted transistors to mitigate the reduction in output impedance in a single transistor, which was inherent to the halo-implants. Furthermore, a more detailed discussion about an array of unity halo-implanted transistors can be found in [10,21,22].

All unity pMOS and nMOS transistors inside the distributed layout had their aspect ratio (W/L) equal to (2.0-μm/2.0-μm) and (0.4-μm/0.6-μm), enabling threshold voltages of 230-mV and 190-mV, respectively.

In addition, we performed a parallel association of six p-MOS and three nMOS to maintain a weak inversion operation, matching the CMOS inverter threshold to $V_{DD}/2$ for a 0.25-V power supply, and accomplishing an overall reduction of the random offset.

The basic CMOS inverter cell had a threshold voltage ($V_{TH}$) of 125-mV and a 35-nA short circuit current ($I_{SC}$), as discussed in Section 2.3, and illustrated in Figure 3.

### 4.1. Simulated Results

The proposed FDDTA was simulated in the Spectre simulator with BSIM models and implemented in the GF 130-nm CMOS process. Table 1 contains the values extracted through computer simulation for pMOS and nMOS transistors inside the distributed layout.

**Table 1.** Parameter for pMOS and nMOS transistors inside the distributed layout.

| Parameter | Value |
|:---:|:---:|
| $g_{o_p}$ | 9.46-n$\Omega^{-1}$ |
| $g_{o_n}$ | 9.45-n$\Omega^{-1}$ |
| $n$ | 1.26 |

The Figures 6 and 7, display the results of slew-rate response, simulated for a 0.5-mV differential input within a buffer configuration, using three different load capacitances $C_L$ of 15 pF, 30 pF and 60 pF on each output. As can be observed in the figure, the delay of the output stayed under 10% and, hence, it could be concluded that the circuit provided a fast response to AC signals.

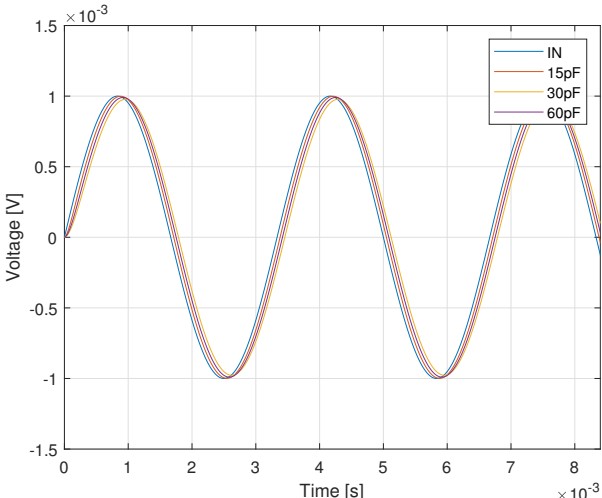

**Figure 6.** Slew rate.

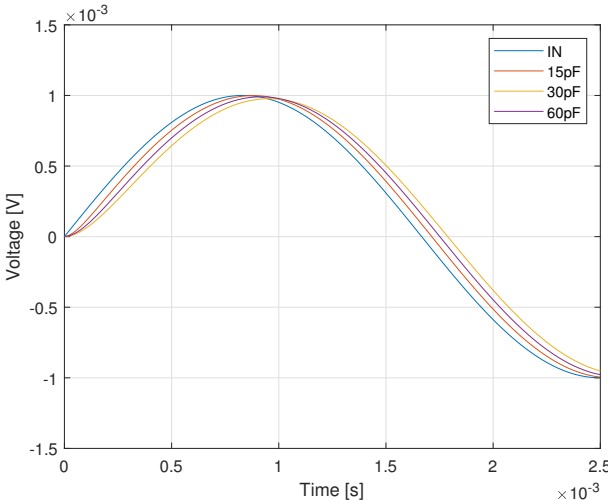

**Figure 7.** Slew rate with zoom.

Figure 8 presents the simulation of the FDDTA step response by applying a differential pulse $V_{in}$ of 10 m$V_{pk}$, with an output load $C_L$ of 15 pF, in both outputs, and evaluating the FDDTA response. We could observe the response behavior of a first-order circuit, as depicted in Equation (12), with a time constant $\tau = (C_L + C_p)/(4G_o)$ and a rise time equal to $T_{rise}(90\%) = 4.57$ ms.

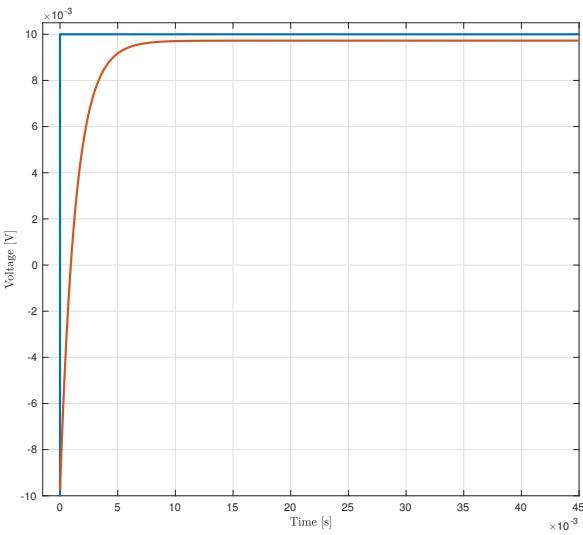

**Figure 8.** Step response for the FDDTA.

Figure 9 presents the simulation of the differential transconductance of the FDDTA structure by sweeping the $V_{id1}$ and $V_{id2}$ from $-125$ mV to 125 mv ($V_{id}/2$) and evaluating the output current $\partial I_{od}/\partial V_{id}$ when $V_{id1} = V_{id2} = 0$.

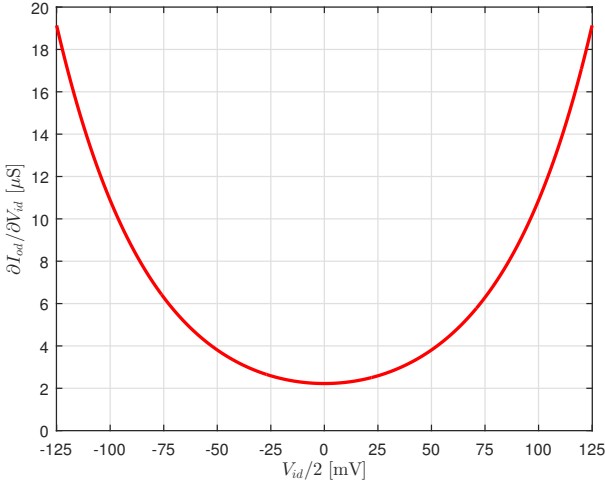

**Figure 9.** Differential transconductance $\partial I_{od}/\partial V_{id}$.

By invoking (9), and using the definitions presented in [10] we can obtain

$$\frac{\partial I_{od}}{\partial V_{id}} = q\frac{I_{SC}}{kT}\left[\frac{1}{n_p}\cosh\left(q\frac{V_{id}}{2n_pkT}\right) + \frac{1}{n_n}\cosh\left(q\frac{V_{id}}{2n_nkT}\right)\right]. \tag{13a}$$

and transconductance of FDDTA, $G_m^{FDDTA}$, when $V_{id1} = 0$ and $V_{id2} = 0$, is defined by

$$G_m^{\text{FDDTA}} = q\frac{2I_{SC}}{nkT} = \frac{70\text{n}}{1.26 \times 25.9\text{m}} = 2.22\mu S, \tag{13b}$$

which was very close to the simulated value of 2.26 µS, as shown in Figure 9.

Figure 10 shows the open-loop magnitude and phase characteristics of the FDDTA with a load capacitance of 30 pF in each output. The proposed circuit offered a gain magnitude around 28 dB, with a cut-off frequency of around 480 Hz, and the gain $A_0$ was highly sensitive to the transistors' mismatch. As expressed in (7), and also in Figure 11,

we can see the results of a Monte Carlo simulation with 1000 samples that followed a normal distribution and μ of 27.78 dB and which, moreover, shows that the distributed layout/schematic technique intrinsically matches the CMOS inverters, maintaining the circuit under accurate control.

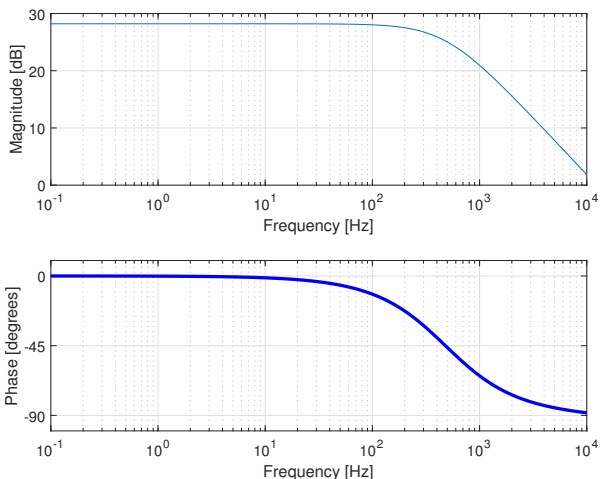

**Figure 10.** Open loop gain and phase.

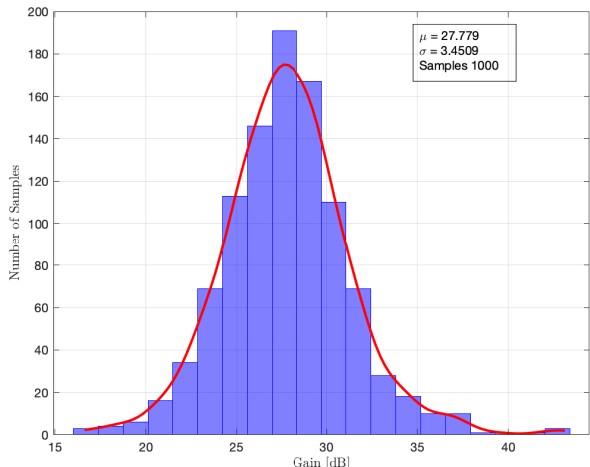

**Figure 11.** Monte Carlo simulation of open loop gain.

Taking (12), we obtain the analytical open loop gain, and compare the result with Figure 10

$$A_o^{\text{FDDTA}} = \frac{[v_{o_p}(s) - v_{o_n}(s)]}{[v_{pp}(s) - v_{pn}(s)] - [v_{np}(s) - v_{nn}(s)]} = \frac{1}{4} \frac{g_{m_p} + g_{m_n}}{g_{o_p} + g_{o_n}} \tag{14a}$$

$$A_o^{\text{FDDTA}} = \frac{1}{4} \frac{g_{m_p} + g_{m_n}}{g_{o_p} + g_{o_n}} = \frac{1}{4} \frac{2.26\mu}{9.46\text{n} + 9.45\text{n}} = 29.80, \tag{14b}$$

this can be expressed in decibels as 29.4 dB, which was very close to the simulated value of 28.2 dB.

The CMRR and PSRR at low frequencies were 54.98 dB and 37.52 dB, respectively, shown in Figures 12 and 13, followed by their respective Monte Carlo simulations (Figures 14 and 15), also show the circuit was under accurate control, provided by the distributed layout/schematic technique. The simulated THD was 1.09% with 0.5-Hz resolution output spectrum for a common mode level of 125-mV, with a differential sinusoidal wave of 175-m$V_{pp}$@100-Hz. For this configuration the dynamic range was 40.52 dB.

In Tables 2–4 the PVT corners of the proposed circuit are, respectively, shown. The MOS transistor corners were slow–slow (SS), slow–fast (SF), fast–slow (FS) and fast–fast (FF),the voltage corners were ±10% and the temperature corners were −20 °C and 100 °C. Therefore, we could conclude that the proposed circuit had acceptable on-chip integration.

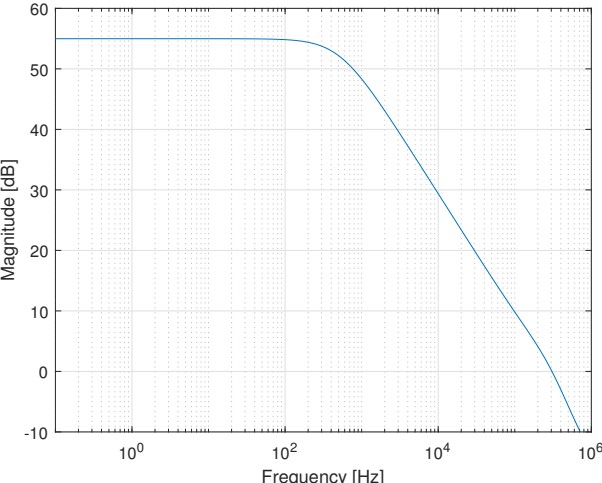

**Figure 12.** CMRR.

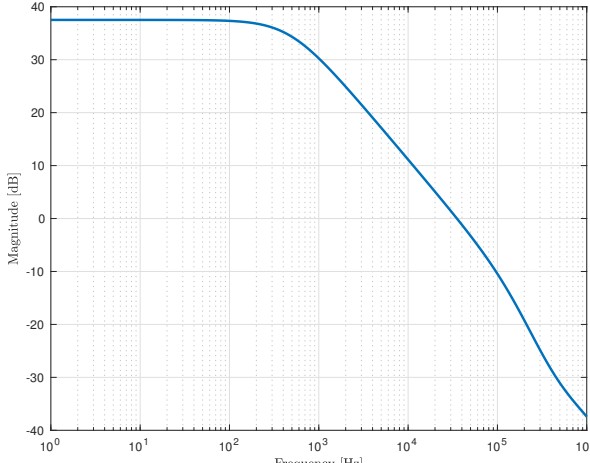

**Figure 13.** PSRR.

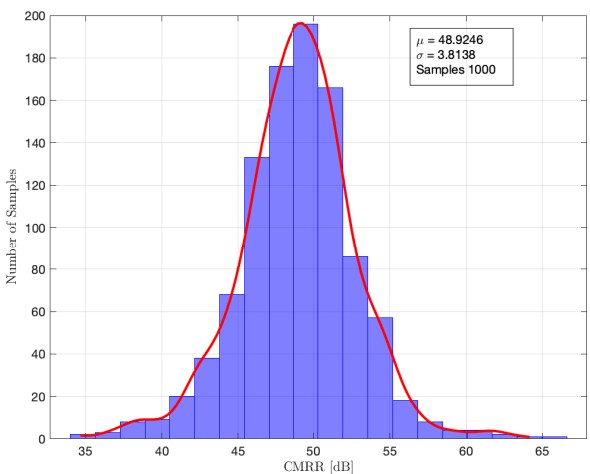

**Figure 14.** Monte Carlo simulation of CMRR.

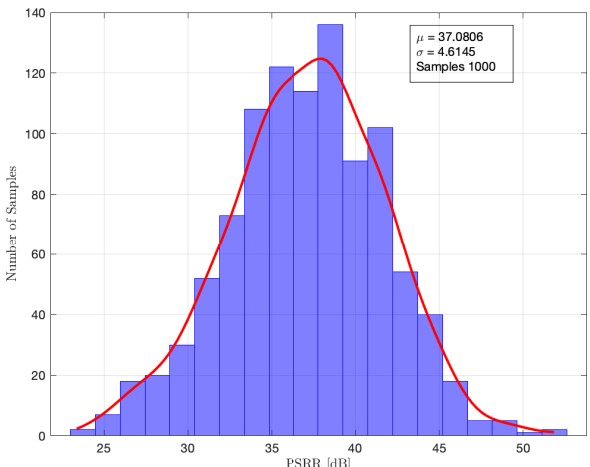

**Figure 15.** Monte Carlo simulation of PSRR.

**Table 2.** Process Corners.

|  | **SS** | **SF** | **TT** | **FS** | **FF** |
|---|---|---|---|---|---|
| Gain (dB) | 22.11 | 24.89 | 28.20 | 27.08 | 29.31 |
| GBW (Hz) | 308.40 | 401.19 | 479.75 | 298.57 | 515.62 |
| CMRR (dB) | 51.97 | 56.12 | 54.98 | 53.43 | 58.07 |
| PSRR (dB) | 34.85 | 21.97 | 37.52 | 42.22 | 38.09 |

**Table 3.** Temp. Corners.

|  | **SS** | | | **TT** | | | **FF** | | |
|---|---|---|---|---|---|---|---|---|---|
| Temp | −20 | 27 | 100 | −20 | 27 | 100 | −20 | 27 | 100 |
| Gain (dB) | 29.43 | 29.33 | 28.34 | 19.33 | 28.20 | 25.34 | 26.15 | 27.31 | 25.99 |
| GBW (Hz) | 479.19 | 480.15 | 481.20 | 451.19 | 479.75 | 471.20 | 430.28 | 464.42 | 480.00 |
| CMRR (dB) | 54.20 | 55.01 | 56.96 | 51.28 | 54.98 | 58.36 | 55.28 | 52.57 | 58.36 |
| PSRR (dB) | 35.96 | 34.85 | 27.27 | 39.24 | 37.52 | 26.52 | 41.23 | 38.09 | 24.21 |

**Table 4.** Voltage Corners.

| VDD (mV) | 225 | 250 | 275 |
|:---:|:---:|:---:|:---:|
| Gain (dB) | 27.45 | 28.20 | 31.36 |
| GBW (Hz) | 469.32 | 479.75 | 480.98 |
| CMRR (dB) | 53.01 | 54.98 | 55.30 |
| PSRR (dB) | 28.98 | 37.52 | 38.35 |

*4.2. Measured Results*

We performed the measurements in a fully differential buffer configuration of the proposed FDDTA. This configuration enabled us to analyze the compatibility between input and output swing, according to Figure 2.

The measurement setup included a Semiconductor Analyzer B1500A and a Dynamic Signal Analyzer DSA35670A, both operating at room temperature (27 °C). In addition, the load capacitance was 30-pF to each output pin. Figure 16 shows the micrography of the test chip.

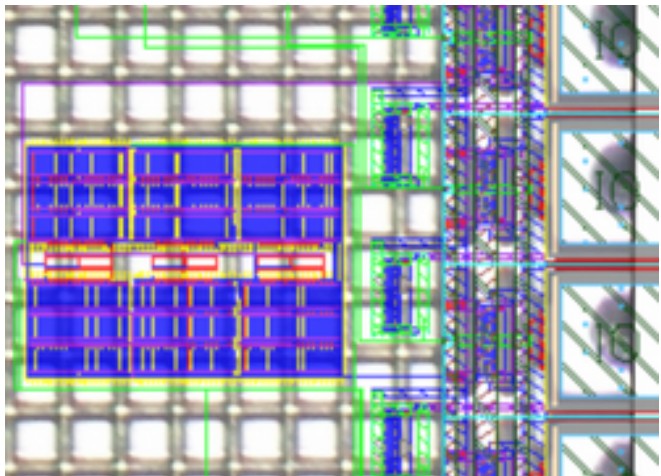

**Figure 16.** Circuit micrograph overlayed with the layout.

Figure 17 shows the measured output, and the input signals, for a differential sinusoidal wave of 100-Hz with an amplitude of 175-mV peak-to-peak, applied to the FDDTA inputs. It shows that the FDDTA differential output replied to the differential input signal with some reduction in the output range.

Furthermore, Figure 18 shows the measured Bode plot for the proposed FDDTA buffer configuration with a cut-off frequency of 3.2-kHz, and. therefore, highlights the first-order system behavior of the fully-differential buffer configuration.

We measured the harmonic distortion, depicted in Figure 19, using the DSA35670A Dynamic Signal Analyzer. For instance, we applied, to the FDDTA inputs, a common mode level of 125-mV with a differential sinusoidal wave of 175-m$V_{pp}$@100-Hz, while the DSA35670A was set up with a 100-kHz sample frequency that resulted in a 0.5-Hz FFT resolution. For this scenario, we expected a 1% $HD_3$ and a $HD_2$ with a small and controlled amplitude, leading to a THD $\approx HD_3$, exactly as depicted in Figure 19. In summary, all those measurements led us to endorse the proposed FDDTA as being fully functional in accordance with the developed models.

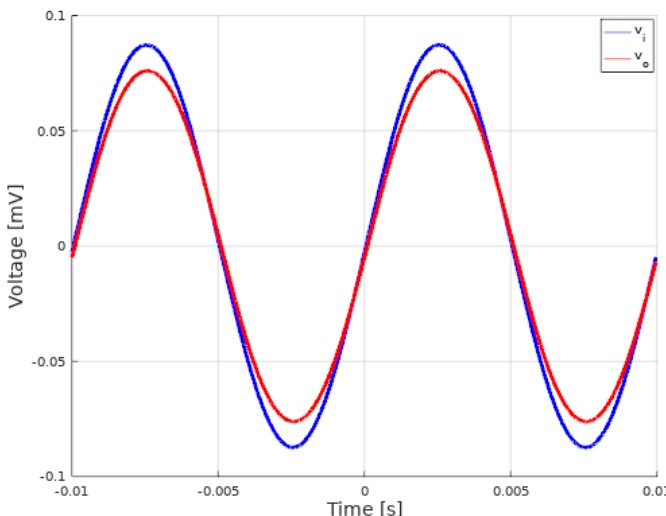

**Figure 17.** Fully-differential buffer configuration: measured input and output signals, for a differential sinusoidal wave of 100-Hz with an amplitude of 175-mV peak-to-peak, applied to the FDDTA inputs.

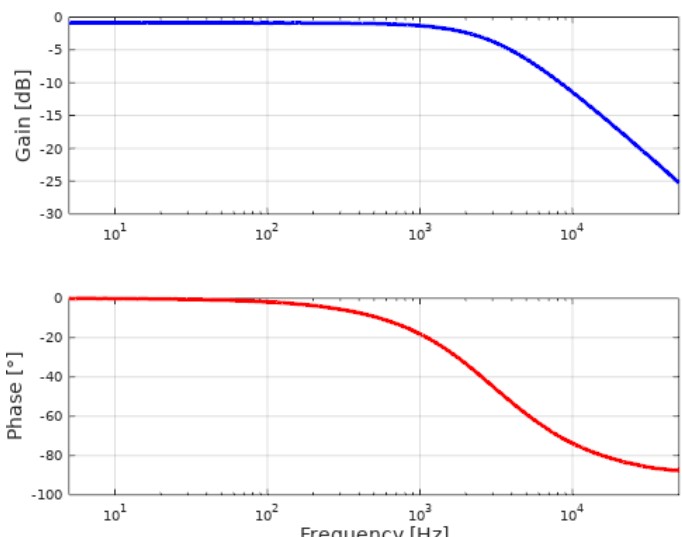

**Figure 18.** Fully-differential buffer configuration: measured frequency response with a cut-off frequency of 3.2-kHz.

Table 5 shows a performance comparison between this work and other low-voltage and low-power FDDTAs, where our proposed architecture featured the smallest supply voltage of 0.25V and the linearity of the proposed circuit was consistent with the other works.

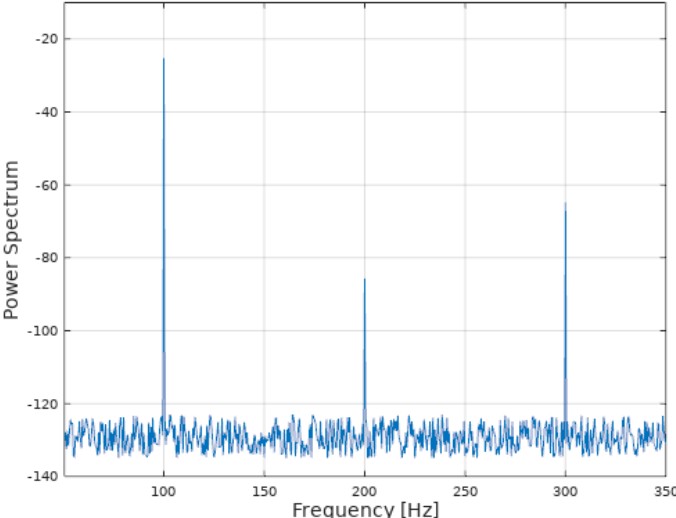

**Figure 19.** Fully-differential buffer configuration: measured harmonic distortion. The 0.5-Hz resolution output spectrum for a common mode level of 125-mV with a differential sinusoidal wave of 175-m$V_{pp}$@100-Hz, leading to a THD $\approx$ HD$_3$.

**Table 5.** Performance comparison between proposed FDDTA and other low-voltage low-pass FD-DTAs architectures.

| Parameters | This Work | IEEE Access 2022 [23] | Sensors 2022 [24] | IEEE TCAS I 2018 [25] | IEEE 2015 [26] | IEEE 2015 [27] |
|---|---|---|---|---|---|---|
| Technology | 0.13 µm | 0.18 µm | 0.18 µm | 0.18 µm | 0.18 µm | 0.5 µm |
| Supply voltage | 0.25 V | 0.5 V | 1.2 V (±0.6 V) | 0.3 V | ±0.4 V | ±2 V |
| Gain | 28.20 dB | 93 dB | - | 60 dB | 1-20 dB | - |
| Transconductance | 2.26 µS | 10.7 nS | 66 µS | 67.7 nS | - | 24 µS to 468 µS |
| −3 dB bandwidth | 480 Hz | <1 Hz | 6.4 MHz | <10 Hz | 23 MHz | 1 GHz |
| Output conductance | 18.91 nS | - | - | - | 111 nS | - |
| Power consumption | 75.30 nW | 205.5 nW | 6 µW | 22 nW | 20 µW | l.66 mW |
| CMRR | 54.98 dB | 67.19 dB | - | 82 dB | - | - |
| PSRR | 37.52 dB | 81.52 dB | - | 57 dB | - | - |
| GBW | 479.75 Hz | 18.02 kHz | - | 1.85 kHz | - | - |
| DR | 40.52 dB | 49.7 dB | 63.59 dB | 57 dB | - | - |

## 5. Conclusions

This paper introduced a fully-differential difference transconductance amplifier architecture, based on CMOS inverters. This design employed an array of halo-implanted MOS transistors to reduce the negative effects of halo implants on output impedance and better match the CMOS inverters.

The circuit was implemented in a 130-nm CMOS process and operated in weak inversion for a 0.25-V power supply; thereby accomplishing specifications suitable for low-frequency applications.

The measurement results. in accordance with the developed theory, endorsed our proposed architecture, based on CMOS inverters. In fact, it spared supplementary external calibration circuits, while keeping performance.

**Author Contributions:** Conceptualization, O.S.S. and R.A.d.S.B.; methodology, R.A.d.S.B., P.M.P., L.H.d.C.F. and G.D.C.; measurement support, P.M.P.; investigation, O.S.S., P.M.P. and R.A.d.S.B.; data curation, O.S.S. and P.M.P.; writing—original draft preparation, O.S.S., P.M.P. and R.A.d.S.B.; writing—review and editing, L.H.d.C.F. and G.D.C.; funding acquisition, R.A.d.S.B. All authors have read and agreed to the published version of the manuscript.

**Funding:** This work was supported, in part, by the Brazilian National Council for Scientific and Technological Development (PQ 303090/2018-9 and GD 140929/2017-7) and FAPEMIG. The authors

would like to thank MOSIS for the chip fabrication. The APC was funded by Federal University Itajuba and Institute of Science and Technology—ICT Unifei Itabira.

**Data Availability Statement:** Not applicable.

**Conflicts of Interest:** The authors declare no conflict of interest.

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
