# Peer review of "A Fully Differential Difference Transconductance Amplifier Topology Based on CMOS Inverters"

_electronics, doi:10.3390/electronics12040963_

Round 1

Reviewer 1 Report

Authors present a fully differential difference transconductance amplifier (FDDTA) architecture based on CMOS inverters. It is designed in a 130-nm CMOS process it operates in weak inversion when supplied with 0.25-V.7

Being it relies on the distributed layout technique that intrinsically matches the CMOS inverters, the FDDTA requires no supplementary external calibration circuit like tail current or bias voltage sources.

The topic has a good novelty degree, inverter-based amplifiers or DigOTAs are circuital blocks which are istigating a certain interesting in the community of electronics engineers and researchers. The paper is well-structured but needs to be further inproved in the contains. My comments are in the following.

1) Being a inverter-based analog block, a lot of works should be referenced, which are missing in this paper. Among them:

- Crovetti, P.S. A Digital-Based Virtual Voltage Reference. IEEE Trans. Circuits Syst. I Regul. Pap. 2015, 62, 1315–1324. 

- Ng, K.A.; Xu, Y.P. A Low-Power, High CMRR Neural Amplifier System Employing CMOS Inverter-Based OTAs With CMFB Through Supply Rails. IEEE J. Solid-State Circuits 2016, 51, 724–737. 

- Toledo, P.; Crovetti, P.; Klimach, H.; Bampi, S.; Aiello, O.; Alioto, M. 300 mV-Supply, sub-nW-Power Digital-Based Operational Transconductance Amplifier. IEEE Trans. Circuits Syst. II Express Briefs 2021, 5, 1.

- Ballo, A.; Pennisi, S.; Scotti, G. 0.5 V CMOS Inverter-Based Transconductance Amplifier with Quiescent Current Control. J. Low Power Electron. Appl. 2021, 11, 37. https://doi.org/10.3390/jlpea11040037

- Bae, W. CMOS inverter as analog circuit: An overview. J. Low Power Electron. Appl. 2019, 9, 26.

- Palani, R.K.; Harjani, R. Inverter-Based Circuit Design Techniques for Low Supply Voltages; Springer: Berlin/Heidelberg, Germany, 2018.

- Ballo, A, Grasso, AD, Pennisi, S. Active load with cross-coupled bulk for high-gain high-CMRR nanometer CMOS differential stages. Int J Circ Theor Appl. 2019; 47: 1700– 1704. https://doi.org/10.1002/cta.2684

- Grasso, A.D.; Pennisi, S.; Scotti, G.; Trifiletti, A. 0.9-V Class-AB Miller OTA in 0.35-µm CMOS With Threshold-Lowered Non-Tailed Differential Pair. IEEE Trans. Circuits Syst. I Regul. Pap. 2017, 64, 1740–1747.

2) Monte Carlo simulations (Gain, CMRR, PSRR) and step response are missing among the simulation results. It should be provided because rejection parameters, as CMRR and PSRR, are very sensible to mismatches among the various paths and cells in the inverter-based topologies. Corner simualtions, which act as common mode variations, are surely useful but not sufficient to evaluate the robustness of the proposed circuit to PVT.

3) What is the dynamic range of the proposed amplifier? 

4) Step response and results (or a table with statistical results) about more than one sample are missing among the measurements. A single case is not enough in this case. Please, provide such information.

5) Comparison with the prior art is missing. It should be reported in the paper.

Author Response

Dear Reviewer, answers to your concerns are in blue throughout your comments. We also thank you for the time dedicated to evaluating our manuscript.

_____________________________________________________________________

Authors present a fully differential difference transconductance amplifier (FDDTA) architecture based on CMOS inverters. It is designed in a 130-nm CMOS process it operates in weak inversion when supplied with 0.25-V.

Being it relies on the distributed layout technique that intrinsically matches the CMOS inverters, the FDDTA requires no supplementary external calibration circuit like tail current or bias voltage sources.

The topic has a good novelty degree, inverter-based amplifiers or DigOTAs are circuital blocks that are instigating a certain interest in the community of electronics engineers and researchers. 
Response. Thanks for your encouraging comment. 

The paper is well-structured but needs to be further improved in the contents. My comments are in the following.

Question 1. Being an inverter-based analog block, a lot of works should be referenced, which are missing in this paper. Among them:

- Crovetti, P.S. A Digital-Based Virtual Voltage Reference. IEEE Trans. Circuits Syst. I Regul. Pap. 2015, 62, 1315–1324. esse artigo aqui não tem nada relacionado com o trabalho

- Ng, K.A.; Xu, Y.P. A Low-Power, High CMRR Neural Amplifier System Employing CMOS Inverter-Based OTAs With CMFB Through Supply Rails. IEEE J. Solid-State Circuits 2016, 51, 724–737. 

- Toledo, P.; Crovetti, P.; Klimach, H.; Bampi, S.; Aiello, O.; Alioto, M. 300 mV-Supply, sub-nW-Power Digital-Based Operational Transconductance Amplifier. IEEE Trans. Circuits Syst. II Express Briefs 2021, 5, 1.

- Ballo, A.; Pennisi, S.; Scotti, G. 0.5 V CMOS Inverter-Based Transconductance Amplifier with Quiescent Current Control. J. Low Power Electron. Appl. 2021, 11, 37. https://doi.org/10.3390/jlpea11040037

- Bae, W. CMOS inverter as analog circuit: An overview. J. Low Power Electron. Appl. 2019, 9, 26. nem sei como um negocio desse foi aceito ta parecendo relatorio de laboratorio da graduacao não tem onde colocar

- Palani, R.K.; Harjani, R. Inverter-Based Circuit Design Techniques for Low Supply Voltages; Springer: Berlin/Heidelberg, Germany, 2018.

- Ballo, A, Grasso, AD, Pennisi, S. Active load with cross-coupled bulk for high-gain high-CMRR nanometer CMOS differential stages. Int J Circ Theor Appl. 2019; 47: 1700– 1704. https://doi.org/10.1002/cta.2684 outro relatorio de faculdade

- Grasso, A.D.; Pennisi, S.; Scotti, G.; Trifiletti, A. 0.9-V Class-AB Miller OTA in 0.35-µm CMOS With Threshold-Lowered Non-Tailed Differential Pair. IEEE Trans. Circuits Syst. I Regul. Pap. 2017, 64, 1740–1747.

Response 1: Thank you for the comment about the manuscript. Attending to your request, we have included the suggested works of high quality and relevant background in the context of our paper. The main idea of each of them was inserted in our manuscript with the reference (lines 16-17, 21-22,34-36, 41-43). We appreciate your contribution.

Question 2. Monte Carlo simulations (Gain, CMRR, PSRR) and step response are missing among the simulation results. It should be provided because rejection parameters, such as CMRR and PSRR, are very sensitive to mismatches among the various paths and cells in the inverter-based topologies. Corner simulations, which act as common mode variations, are surely useful but not sufficient to evaluate the robustness of the proposed circuit to PVT.

Response 2.  Thank you for this precious observation. We have included Figures 11, 13 and 15 to address this issue and the discussion text in lines 184-186, 189-193.

Question 3. What is the dynamic range of the proposed amplifier?

Response 3. Thank you for your important suggestion. We have included the DR value in lines 192-193 and in the comparison table.  

Question 4. Step response and results (or a table with statistical results) about more than one sample are missing among the measurements. 

Response 4. Dear reviewer, we fully agree with your request: more than just one sample is required for measurements. However, this chip was sent to manufacture only 40 units. These 40 units were the maximum limit of the MOSIS agreement and were manufactured in 2016 in the GF_8RF process. Weak inversion circuits were measured over 2016-2019 (including those in this manuscript). These units contained other students' projects; after handling and measurements, none of them are currently working. We apologize for this.

However, the Monte Carlo tests and process corners were inserted in the manuscript as requested, as well as the step response (lines 171-173 and figure 8). The simulations plotted together with the measured, and calculated curves can prove the accuracy of the model (BSIMv4) and the robustness of the proposed circuit. I hope this can satisfies your concern.

Question 5. A comparison with the prior art is missing. It should be reported in the paper.

Response 5. Thank you for bringing our attention to this issue. For comparison, we have included a comparison table with other low-voltage and low-power FDDTAs architectures operating in different BWs and supplies voltages. We would like to inform you that our proposed architecture features the smallest supply voltage of 0.25V, and the linearity of the proposed circuit is consistent with the other reported works. Inserted in Table 5 and discussion in lines 221-223.

Reviewer 2 Report

For a better understanding of the material, I would suggest the Authors would address the following MINOR points:

A.      In the text there is not a clear comparison between the results obtained by the analytical formulation and the measurements. The Authors are invited to introduce this comparison in form of tables, graphs with multiple curves.

B.      The authors are also invited to comment the tables/graphs above offering some consideration on the accuracy and efficiency of their proposed design.

Author Response

Dear Reviewer, answers to your concerns are in blue throughout your comments. We also thank you for the time you dedicated to evaluating our manuscript.

______________________________________________________________

For a better understanding of the material, I would suggest the Authors would address the following MINOR points:

Question 1.In the text there is not a clear comparison between the results obtained by the analytical formulation and the measurements. The Authors are invited to introduce this comparison in form of tables, graphs with multiple curves.

Response 1. Thank you for this precious observation. We addressed these analytical formulations and the measurements in Equations 13b and 14b for the transconductance, output conductance, and open-loop gain parameters.

Question 2. The authors are also invited to comment the tables/graphs above offering some consideration on the accuracy and efficiency of their proposed design.

Response 2. Thanks for your suggestion. We have added Monte Carlo simulations for the gain, CMRR, PSRR, and step response parameters. These simulations can prove the accuracy and efficiency of the proposed design against mismatch issues. Please verify these changes in the revised manuscript; we hope these changes can satisfy your concerns.

Reviewer 3 Report

It is 32% of plagiarism

In this form I recommended no publication for this paper

Round 2

Reviewer 1 Report

Authors have satisfactorily answered to all my raised concerns and comments.

Reviewer 3 Report

thank you for all authors for this rectification